# Effect of *Vachellia tortilis* Leaf Meal and Sunflower Oil Inclusion in Supplementary Diets of Lambs on In Vitro Short-Chain Fatty Acid and Gas Production and In Vivo Growth Performance

**DOI:** 10.3390/ani15060863

**Published:** 2025-03-17

**Authors:** Mahlogonolo Daniel Serumula, Bulelani Nangamso Pepeta, Mehluli Moyo, Terence Nkwanwir Suinyuy, Ignatius Verla Nsahlai

**Affiliations:** 1Animal and Poultry Science, School of Agricultural, Earth and Environmental Sciences, University of KwaZulu-Natal, Private Bag X01, Scottsville 3209, South Africa; mahlogonolod5@gmail.com (M.D.S.); bulelani.pepeta@gmail.com (B.N.P.); mehluli.moyo@live.com (M.M.); 2Faculty of Natural and Agricultural Sciences, Department of Animal Sciences, University of Pretoria, Private Bag X20, Pretoria 0002, South Africa; 3School of Biology and Environmental Sciences, University of Mpumalanga, Private Bag X11283, Mbombela 1200, South Africa; tsuinyuy@gmail.com

**Keywords:** condensed tannins, forage legumes, gas production, polyunsaturated fatty acids

## Abstract

This study investigated the potential effects of *Vachellia tortilis* leaf meal (VT) and sunflower oil (SFO) supplementation on in vitro short-chain fatty acid (SCFA) production and gas production and the in vivo growth performance of lambs. The VT inclusion showed no statistically observed negative impact on growth performance or in vitro gas production, suggesting that VT might be a suitable supplementary protein source for lambs in the subtropical regions, given that further validation studies conform with the findings from the current study.

## 1. Introduction

The productivity of grazing ruminants is mostly affected by seasonal fluctuations in the quality and quantity of available forages. This is due to pasture quality being lowest during the dry season in the tropical and subtropical regions of Southern Africa [1]. During the dry season, ruminants in tropical regions generally graze on mineral and nitrogen-deficient grasses, rendering their nutritional value as poor. These low nutritional value grasses are poorly digested in the rumen, producing large quantities of enteric methane upon digestion. As a result, ruminants contribute to global warming through the production of enteric methane gas as a by-product of fermentation of structural carbohydrates in the rumen. Hence, reducing the carbon footprint of ruminants for sustainable ruminant production has necessitated the exploration of methane-mitigating strategies, including improvement in the nutritional value of ruminant diets [2].

One major strategy used to improve the nutritional value of ruminant diets is the supplementation of roughage-based diets with grain-based concentrates [3,4]. This practice is widely used in intensive dairy farming systems and has the desired outcome of reduced methane yields [5,6]. However, there are setbacks regarding the use of such dietary manipulation strategies for ruminant livestock kept in extensive communal production systems. In these systems, ruminants depend primarily on grazing on veld grass as a major way of obtaining nutrients, with little or no concentrate supplementation [7,8]. Resource-limited communal farmers in tropical and subtropical regions of Sub-Saharan Africa barely supplement grazing livestock with cereal and legume grain-based concentrates. This is partly due to the high cost of concentrates that form part of the food resources for humans. Additionally, most of these regions experience erratic rainfall patterns, rendering them unsuitable for large-scale crop production for human consumption and livestock feeding. Hence, it is of prime interest to develop dietary strategies aimed at reducing the inclusion of grains on ruminant diets while at least maintaining enteric methane gas emissions. Such strategies give an impression of being easily adopted by farmers in the subtropical regions. Use of cheap alternative non-conventional feed resources as ingredients in concentrate formulations can serve as a valuable means to reduce the inclusion of grains in ruminant supplementary diets. Leguminous multipurpose tree leaves can replace small proportions of grains in concentrate supplements in conjunction with urea-treated basal roughage-based forages such as veld hay to improve the nutritional value of ruminant diets.

The use of tannin-containing plants, including *Vachellia tortilis*, as dietary supplements in ruminant diets has been reported to modulate rumen fermentation patterns and reduce methane emissions [9,10]. Similarly, polyunsaturated fatty acids such as sunflower oil have shown promise in reducing methane emissions by shifting fermentation patterns and decreasing protozoa populations [11,12]. However, the effectiveness of *Vachellia tortilis* leaf meal and sunflower oil, and their combined inclusion in ruminant diets to modulate fermentation patterns and reduce methane emissions, is relatively undocumented. It was hypothesized that the inclusion of *Vachellia tortilis* leaf meal and sunflower oil in concentrate diets would decrease the in vitro acetate/propionate ratio and methane and carbon dioxide production while maintaining the growth performance of lambs. Therefore, the objective of this study was to explore the potential effects of including *Vachellia tortilis* leaf meal and sunflower oil in concentrate diets on in vitro short-chain fatty acids, methane and carbon dioxide production, and the growth performance of lambs.

## 2. Materials and Methods

### 2.1. Study Site and Ethics

The experiment was conducted at Ukulinga research farm at the University of KwaZulu-Natal, Pietermaritzburg, South Africa. The area is located at 29°24′ E and 30°24′ S in the subtropical hinterland, at an altitude of approximately 700 m above sea level. The area receives an annual rainfall of 735 mm, which falls between October and April, with average minimum and maximum temperatures of 8.9 °C and 25.7 °C, respectively. The characteristic vegetation involves various tree and grass species that include *Vachellia karroo* (sweet thorn), *Vachellia nilotica* (thorn mimosa), *Vachellia sieberiana* (paperbark thorn), *Themedia trianda* (kangaroo grass), and *Heteropogon contortus* (black spear grass). The experimental protocol was approved by the Animal Ethics Committee of the University of KwaZulu-Natal with the reference number AREC/057/017M on research conducted in animals following international principles for animal use and care.

### 2.2. Leaf Meal Collection and Preparation

*Vachellia tortilis* leaves were harvested at Makhathini Research Station, Jozini in KwaZulu-Natal province located at 27°43′ S and 32°14′ E geological coordinates, South Africa. *Vachellia tortilis* was selected based on its nutritive value (crude protein content) and presence of phenolic compounds (i.e., tannins). Leaves were harvested during the post-rainy season at an advanced stage of maturity and were dried separately under shade for three days and sieved to eliminate thorns, pods, and twigs. Thereafter, leaves were ground to pass through a 2 mm sieve using a hammer mill (Scientec hammer mill 400, Lab World Pty Ltd., Johannesburg, South Africa) to produce the leaf meal. The inclusion of *Vachellia tortilis* leaf meal in ruminant diets is justified by its wide availability, nutritional value, and potential to serve as a sustainable protein source in regions where conventional feed resources are scarce. Vachellia tortilis is indigenous to various regions in Sub-Saharan Africa, including South Africa, particularly in the KwaZulu-Natal, Limpopo, and North-West provinces [13]. Its ability to thrive in arid and semi-arid environments makes it a valuable feed resource, especially in areas prone to drought and limited pasture productivity.

The leaves of *Vachellia tortilis* are rich in crude protein (CP), ranging between 11 and 19%, with relatively low neutral detergent fiber (NDF) (35–55%) and acid detergent fiber (ADF) (20–40%) contents [13]. These chemical properties make *Vachellia tortilis* leaf meal a suitable supplement to improve the nutritional quality of roughage-based diets for ruminants, particularly in communal farming systems.

Furthermore, the tannins in *Vachellia tortilis* leaves, while known for their potential anti-nutritional effects, can play a beneficial role in modulating ruminal fermentation and reducing methane emissions [14]. This aligns with the growing interest in using tannin-rich plants to improve livestock productivity and reduce greenhouse gas (GHG) emissions in tropical and subtropical production systems [15,16,17].

Therefore, the incorporation of *Vachellia tortilis* leaf meal in ruminant diets presents an affordable and environmentally friendly feeding strategy to support sustainable livestock production in resource-limited regions of Sub-Saharan Africa.

### 2.3. Experimental Design, Experimental Diets, and Growth Performance

The concentrate diets were formulated to meet animal total energy and protein requirements for maintenance, growth, and wool production according to the methods of [18]. To determine the effect of *Vachellia tortilis* leaf meal and sunflower oil inclusion in supplement diets on in vitro fermentation products and growth performance of lambs fed *Themeda trianda*-based diets, four dietary treatments were formulated (Table 1). Concentrate diets were fed at a restriction feeding level of 480 g/day per head of lamb, while urea-treated *Themeda trianda* hay was offered ad libitum as a basal diet, with free access to water. The diets were as follows: (i) control diet (CL), (ii) *Vachellia tortilis* leaf meal diet (VT), (iii) sunflower oil diet (SFO), and (iv) combination of *Vachellia tortilis* leaf meal/sunflower oil diet (VSFO).

Eight merino lambs (mean initial body weight: 27.15 ± 3.0 kg) in two live weight blocks (four and four) were randomly assigned into four treatments within each block, making two animals per treatment. This experimental design aligned with the studies of the authors of [19,20,21,22]. During period two and three, the animals were reassigned to treatments, making sure that no animal went to its previous treatment. The experiment lasted for 126 days, with three periods each lasting for 42 days. At the beginning of each period, the lambs were allowed to adapt to dietary treatments and the pens for 14 days, followed by 28 days of data collection. Starting from days 19–21 in each period, the lambs were adapted to carrying fecal bags, followed by 7 days of total fecal collection. Daily, fecal bags were emptied in the morning before feeding. Feed offered, refusals, spillage, and fecal samples were oven dried at 60 °C for 72 h and weighed. For each animal, dry matter intake (DMI) was calculated as the amount of feed DM offered minus dry refusal and spillages. Apparent digestibility (APD) was calculated thus:(1)$$\mathrm{APD}\;\left(g/\mathrm{kg}\right)=\frac{\mathrm{total}\;\mathrm{dry}\;\mathrm{matter}\;\mathrm{intake}\;\left(\mathrm{DM}\right)-\;\mathrm{total}\;\mathrm{faecel}\;\mathrm{output}\;\left(\mathrm{DM}\right)}{\mathrm{total}\;\mathrm{dry}\;\mathrm{matter}\;\mathrm{intake}\;\left(\mathrm{DM}\right)}\times 1000$$

During each period, the lambs were weighed for two successive days at the beginning and at the end, and the average was taken as the initial live weight and final live weight. The average daily gain of each animal (ADG) was calculated as the difference between the final and the initial body weight divided by 28 in each period. The feed conversion ratio (FCR) was calculated by dividing the average daily gain (ADG) by the total dry matter intake (DMI).

### 2.4. Chemical Analyses of Dietary Treatments and Fecal Samples

The wet chemistry analyses of the diets (Table 1) and fecal samples were performed using methods described by the Association of Official Analytical Chemists (AOAC, 1990) for dry matter (ID 945.15), ash (ID 942.05), nitrogen (ID 979.09), and crude fat (ID 920.39). The nitrogen content of the diet samples was determined using a Leco Truspec nitrogen (N) analyzer (Leco FP200, LECO, Pretoria, South Africa). Crude protein (CP) was determined by multiplying N content by 6.25. Neutral detergent fiber (NDF), acid detergent fiber (ADF), and acid detergent lignin (ADL) were sequentially determined on the same sample according to Van Soest et al. [23] using an ANKOM 200 fiber analyzer (ANKOM Technology, Fairport, NY, USA). Heat stable amylase and sodium sulfite were used in the NDF assay and results were expressed inclusive of the residual ash. Hemicellulose was estimated as a difference between NDF and ADF. Condensed tannins were analyzed using the butanol/hydrochloric acid (HCl) method according to Reed et al. [24]. Urea-treated hay (UTH) had contents (g/kg DM) of dry matter (815 ± 108), ash (82 ± 4.16), organic matter (76 ± 6.56), CP (85 ± 8.37), NDF (734 ± 13), ADF (488 ± 125), and hemicellulose (335 ± 5.57).

### 2.5. In Vitro Digestion

An incubation system fitted with a temperature controller (OMRON E5F2 Corporation 2007–2018) was used for in vitro digestion and the temperature inside the incubator was maintained at 39 °C. A salivary buffer (Buffer A) was prepared following procedures by Tilley and Terry [25]. Briefly, 19.60 g of NaHCO_3_, 7.40 g of Na_2_HPO_4_, 1.14 g of KCI, 0.94 g of NaCl, and 0.26 g of MgCl_2_ 6H_2_O were dissolved into 2 L of distilled water. Another buffer (Buffer B) was prepared by weighing and dissolving 5.30 g of CaCl_2_ 2H_2_O into 100 mL of distilled water. Two milliliters (2 mL) of buffer B was titrated to buffer A, which was stirred on a magnetic hotplate for 15 min while being bubbled with carbon dioxide until the buffer solution was clear. Approximately 300 mg of concentrate sample was weighed into 250 mL Duran bottles, followed by the addition of 67 mL of the buffer solution. Duran bottles with the mixture of concentrate samples and buffer solution were placed in the incubator so that the temperature in the Duran bottles would equilibrate with the temperature inside the incubator. Rumen fluid was collected from two rumen fistulated Merino wethers that were fed urea-treated *Themeda trianda* hay. Rumen fluid was collected before the morning meal allocation, and strained and filtered through four-layered muslin cloth into a prewarmed (39 °C) thermos flask flushed with carbon dioxide [26]. Rumen fluid was conveyed to the lab and 33 mL was added into each warm Duran bottle while continuously flushing with CO_2_, then sealed with an air-tight lid and quickly placed into the incubation system. Each concentrate sample was replicated four times with four Duran bottles containing rumen fluid, and Duran bottles with only buffer solution were incubated as blanks. The pH readings of the rumen fluid samples were sequentially recorded at time intervals of 2, 4, 16, and 48 h of incubation, and this was repeated 4 times. A Crison micropH 2000 (Crison Instruments SA, Riera Principal, Alella, Spain) was used to measure the pH readings. During each time interval, 5 Duran bottles comprising 4 concentrate diets and 1 blank were removed from the incubator. The pH readings of the bottles were immediately recorded. Then, the Duran bottles were placed in the refrigerator to prevent further fermentation. After cooling, 10 mL syringes were used to collect samples from each Duran bottle for use to determine short-chain fatty acids (SCFAs). Such bottles were centrifuged at 10,000× *g* for 15 min at a temperature range between 4 °C and 8 °C using a Beckman Coulter Centrifuge (Avanti J-26 XPI 6804, Beckman Coulter, Inc., Brea, CA, USA). The supernatant (4 mL) was transferred to 5 mL capped tubes containing 1 mL of 25 % (*w*/*v*) meta-phosphoric acid (H_2_SO_4_) as the internal standard and then refrigerated prior to SCFA analysis.

To determine the in vitro dry matter digestibility (IVDMD), the sediment, after 48 h of digestion, was transferred into gas chromatography-labeled vials and dried for 72 h in the oven at 60 °C. After drying, the vials were cooled in a desiccator and sequentially weighed for in vitro dry matter digestibility (IVDMD) calculation, as follows:(2)$$\mathrm{IVDMD}\;\%\;=\frac{\left[\left(\mathrm{sample}\;\mathrm{mass}\;-\mathrm{undigestible}\;\mathrm{residue}\;\mathrm{mass}\right)-\mathrm{blank}\;\mathrm{mass}\right]\times 100}{\mathrm{sample}\;\mathrm{mass}}$$

### 2.6. Gas Chromatography Analysis of Short-Chain Fatty Acids

Short-chain fatty acids (SCFAs) were analyzed using a Coupled Varian 3800 gas chromatography (Varian, Palo Alto, CA, USA) and Varian 1200 mass spectrometry (GC-MS). The GC-MS was equipped with an Alltech EC-WAX column of 30 m × 0.25 mm internal diameter with 0.25 μm film thickness (Alltech Associates Inc., Deerfield, IL, USA). Helium was used as the carrier gas at a flow rate of 1 mL/min. From each pre-treated sample, 2 μL was injected into a ChromatoProbe trap prepared by cutting glass tubes equaling the size of ChromatoProbe quartz microvials (length: 15 mm; inner diameter: 2 mm) filled with 2 mg of a 50:50 mixture of Tenax TA (Alltech Associates, USA) and graphitized carbon (Carbotrap™, Supelco, Bellefonte, PA, USA), and closed on both ends with glass wool. The ChromatoProbe traps were placed in a Varian 1079 injector by means of a ChromatoProbe fitting and thermally desorbed. The temperature of the injector was initially maintained at 40 °C for 2 min with a 20:1 split ratio and then increased to 200 °C, where it was held for a minute in a splitless mode for thermal desorption.

Compound detection was delayed for 6 min. After a 3 min hold at 40 °C, the GC oven was ramped up to a rate of 10 °C min^−1^ to 240 °C, where it was held for 12 min. Compound identification was carried out using the NIST05 mass spectral library and by comparisons with retention times of chemical standards, along with comparisons between calculated Kovats retention indices and those published in the literature. Clean ChromatoProbe traps were run in GC-MS as controls to identify peak areas and retention times and background contamination. Compounds present at higher or similar percentages to the blanks were considered as contaminants and excluded from the analysis. For quantification of compounds, known amounts of standards of dominant compounds (acetate, propionate, and butyrate) were injected into cartridges and compared with those of the standards and used to calculate the total amount of compound per gram of substrate [27].

### 2.7. Calculation of Carbon Dioxide and Methane Production from the Proportion of Short-Chain Fatty Acids

The proportions of carbon dioxide (CO_2_) and methane (CH_4_) from short-chain fatty acids (SCFAs) were calculated using equations by Wolin [28] and Moss et al. [29] based on stoichiometric laws and the oxidation state of chemical balance and the proportion of fermentation products.

Carbon dioxide (Y) was calculated as follows:(3)$$Y\;=\frac{\mathrm{Ma}}{2}+\frac{\mathrm{Mp}}{4}+3\;\frac{\mathrm{Mb}}{2}$$

Methane (CH_4_) was calculated as follows:(4)$$Z=Y\;–\;\frac{Mp}{2}\;–\;Mb$$
where Y is CO_2_; Z is CH_4_; Ma, Mp, and Mb are molar proportions of acetic acid, propionic acid, and butyric acid, respectively.

On the other hand, the equation by Moss et al. [29]) is solely based on proportions of SCFAs in the rumen fluid to calculate methane (CH_4_) as follows:(5)CH4=0.45C2 – 0.275C3+0.40C4
where C_2_, C_3_, and C_4_ are proportions of acetate, propionate, and butyrate, respectively.

### 2.8. Statistical Analysis

The mixed model procedure (PROC mixed) of SAS version 9.4 was used to determine the effect of *Vachellia tortilis* leaf meal and sunflower oil inclusion in supplements on the in vitro production of SCFAs, methane and carbon dioxide, and growth performance. The initial live weight was used as a covariate. A regression procedure of SAS (PROC REG) was used to determine the precision of the predictions of methane yield from the production of SCFAs using equations by Wolin [28] compared with using the equation by Moss et al. [29]. The probability of difference test was used to separate means that significantly differed from each other at (*p* < 0.05). The statistical model was:(6)$$Yijkl=µ+Di+Pj+Ok+SUB\;\left(O\right)jk\;+ɛijkl$$
where Y*_ijk_* is the observation; µ is the overall mean; D*_i_* is the fixed effect of treatment diet *i* (*I =* 5); P*_j_* is the repeated effect of period *_j_* (*_j_* = 3); O_k_ is the effect of order *k* of assigning treatments to lamb (subjects); SUB (O)*_jk_* is the random effect of subject *j* within order *k*, mean 0, and variance *σ*^2^; *ɛ_ijk_* is the error due to random effects with mean 0 and variance *σ*^2^. A post hoc power analysis was performed to determine whether the sample size was adequate for detecting a meaningful effect given the observed sample size used in the current study.

## 3. Results

### 3.1. Chemical Composition

The chemical composition of the dietary treatments in Table 1 showed significant differences across various components. The *Vachellia tortilis* (VT) leaf meal diet had higher crude protein (CP) content (241 g/kg DM) compared with both the VT and sunflower oil (SFO) diets (*p* = 0.0001). The inclusion of sunflower oil significantly increased the ether extract (EE) content in both the *Vachellia tortilis* leaf meal with sunflower oil (VSFO) and SFO diets, with values of 37.8 g/kg DM and 55.8 g/kg DM, respectively (*p* = 0.0001). Additionally, the VT diet had more condensed tannins (CTs) (7.1 g/kg DM) compared with the VSFO diet (5.6 g/kg DM).

### 3.2. In Vitro Fermentation Parameters

The addition of *Vachellia tortilis* (VT) leaf meal and sunflower oil did not significantly affect total short-chain fatty acids (SCFAs), CH₄, or CO₂ production (Table 2). However, Table 3 depicts that the proportion of butyrate was highest in the VSFO diet (*p* = 0.05), and the VT diet improved in vitro dry matter digestibility (IVDMD) compared with the control diet (*p* = 0.02). Fermentation shifted towards propionic acid production with the inclusion of sunflower oil after 40 h of incubation, while CH₄ production decreased after 30 h. Furthermore, a decreasing trend in ruminal pH was observed in response to VT leaf meal and sunflower oil at different incubation times (2, 4, 16, and 48 h, Figure 1). Strong agreement was found between methane predictions using the Wolin [28] and Moss et al. [29] equations, with a high adjusted R^2^ value of 0.994 (Figure 2). Additionally, in Figure 3, there were curvilinear responses in methane production over hourly incubation times as predicted by both equations, based on stoichiometric ratios in response to the control and sunflower oil inclusion in concentrate rations (*p* < 0.05).

### 3.3. Regression Equations of the Relationship Between Short-Chain Fatty Acid and Methane Production and Length of Incubation Time

Table 4 represents the quadratic regression equations showing that propionate concentration decreased over time, reaching a minimum of 12.5% at 39.5 h, while the acetate-to-propionate (A:P) ratio peaked at 3.53 after 23.3 h, indicating a shift towards acetate-dominant fermentation. Methane production estimated using Moss et al. [29] and Wolin [28] equations, peaked at 32.5% and 37.7% at 29.5 h, respectively, before declining.

### 3.4. Growth Performance

The growth performance of the Merino lambs, as indicated by dry matter intake (DMI), average daily gain (ADG), and feed conversion ratio (FCR), did not show significant differences (*p* > 0.05), except for a reduction in apparent total tract digestibility with sunflower oil inclusion (*p* = 0.03) (Table 5).

## 4. Discussion

The implementation of sustainable sheep production systems in tropical and subtropical regions through the inclusion of multipurpose trees and oils, such as *Vachellia tortilis* and sunflower oil, may improve the nutritional value of diets while maintaining growth performance and mitigating greenhouse gas emissions [3]. In this study, *Vachellia tortilis* leaf meal and sunflower oil provided condensed tannins and ether extracts, which are known to modulate fermentation processes both in vitro and in vivo [30]. Although the inclusion of *Vachellia tortilis* leaf meal increased the crude protein content in diets, it did not significantly affect the proportions of short-chain fatty acids (SCFAs) across dietary treatments (Table 3). This might be due to the lack of changes in the microbial populations in the rumen fluid in response to the increased crude protein content, as reported by Paya et al. [31], which explains why SCFAs were not affected. However, such was not evaluated in the current study, and it is noteworthy to mention that the inferential approach was used to facilitate the discussion of the findings, warranting further studies on the subject matter. The inclusion of sunflower oil increased the ether extract levels in the VSFO and SFO diets, which surprisingly, in turn, increased as opposed to decrease in vitro dry matter digestibility (IVDMD). AS sunflower oil, being a rich source of polyunsaturated fatty acids (PUFAs), has been associated with negative effects on ruminal bacterial populations [32]. However, Camero et al. [33] found that forage legumes such as *Erythrina poeppigiana* and *Gliricidia sepium* had higher potential degradation compared with urea-treated hay, which contributed to relatively high SCFA production. In line with their findings, our results indicated that VT and VSFO diets had a higher IVDMD compared with the control diet.

The proportions of acetate and propionate appeared to counteract each other during incubation, which resulted in no significant changes in ruminal pH, a finding similar to that of Sliwinski et al. [34], who observed no pH changes when plant extracts rich in tannins were added to the rumen of sheep. Moreover, no significant differences were observed in the relationship between hourly incubation time and methane production predicted by both Wolin [28] and Moss et al. [29] equations in response to all dietary treatments. There were also no differences observed in the overall production of acetate (A), propionate (P), the A:P ratio, and methane production. This lack of difference suggests that the inclusion of *Vachellia tortilis* leaf meal in concentrate supplement diets for lambs does not have a detrimental effect on enteric methane emissions. The pH values ranged from 6.4 to 6.8, which is optimal for digestion and microbial activity [8].

In terms of methane production, the regression of SCFAs [28] against the predicted values from the Moss et al. [29] equation showed similar levels of accuracy and precision in estimating methane based on the proportions of SCFAs in the rumen fluid. Therefore, the proportion of methane can be estimated using either equation based on SCFA molar proportions. Similar to the study by Muhlisin et al. [35], our findings showed no effect of *Vachellia tortilis* leaf meal inclusion on supplementary concentrate diets on SCFAs, methane (CH_4_), and carbon dioxide (CO_2_). Similarly, the inclusion of graded levels of 10% and 20% *Leucaena leucocephala* leaf meal had no effect on in vitro fermentation [35]. Although the inclusion of *Vachellia tortilis* leaf meal was expected to reduce methane production due to the presence of condensed tannins, which are generally associated with inhibitory effects on methanogens, the level of *Vachellia tortilis* leaf meal (12.15%) in the diet was too low to have a significant reducing effect on methane production. As a result, *Vachellia tortilis* leaf meal inclusion had similar effects as other treatments on growth, SCFAs, and overall gas production (Table 5), indicating its potential for inclusion in lamb diets.

Condensed tannin-rich *Vachellia tortilis* leaf meal had no effect on dry matter intake (DMI), average daily gain (ADG), feed conversion ratio (FCR), or total tract neutral detergent fiber digestibility (ttNDFd), in line with findings from Animut et al. [36], who reported similar results in lambs fed diets where barley grain was partially replaced with tannin-rich carob pods. Other studies [37,38,39,40] also found no significant differences in dry matter intake in ruminants supplemented with tannin-rich leguminous foliage. This could be attributed to the concentrations of phenolic compounds being far lower than the 50 g/kg DM threshold considered necessary to influence dry matter intake [41]. Consequently, our findings suggest that the condensed tannin content in the VT diet (7.1 g/kg) and VSFO diet (5.6 g/kg) were too low to cause significant changes in DMI, ADG, and FCR. The effects of polyphenolic compounds on sheep performance may vary depending on the source and inclusion levels [42].

Similar to Sutton et al. [43], the inclusion of sunflower oil (40.08 g/kg) in lamb diets did not adversely affect DMI, ADG, or FCR. Lambs fed CL, VT, VSFO, and FSO diets had similar concentrate DMI, suggesting that the inclusion of *Vachellia tortilis*, sunflower oil, and their combination had no detrimental effect on concentrate DMI. Additionally, apparent digestibility was similar between the CL and VT dietary treatments, while the VSFO and FSO diets had lower apparent digestibility compared with the CL and VT diets. The inclusion of oils may coat feed particles, creating a physical barrier that reduces microbial attachment to the digesta in the rumen, which in turn reduces degradability and depresses digestibility [44]. However, the maximum level of *Vachellia tortilis* inclusion in lamb diets that will not induce low feed intake and poor growth performance still needs to be explored. it is noteworthy to clarify that fistulated animals were used to obtain rumen fluid because the study lacked the ethics approval to euthanize the animals for rumen sampling. The use of fistulated animals allowed for the collection of rumen fluid while adhering to ethical guidelines.

## 5. Conclusions

This study explored the effects of 12.15% *Vachellia tortilis* leaf meal and 4.08% sunflower oil, both individually and in combination, on in vitro fermentation parameters, including short-chain fatty acids (SCFAz) and methane (CH_4_) production, in vivo lamb performance. While no significant effects were observed on the in vivo growth performance of lambs, including dry matter intake (DMI), average daily gain (ADG), feed conversion ratio (FCR), and in vitro SCFA production, *Vachellia tortilis* leaf meal increased in vitro dry matter digestibility (IVDMD), and sunflower oil reduced apparent total dry matter digestibility (APD). The use of stochiometric ratios from ruminal formation parameters can be used to predict CH_4_ using either the equations of Wolin [28] or Moss et al. [29], with highly similar degrees of association. *Vachellia tortilis*, being native to the Southern Hemisphere and available in large quantities, gives the impression of offering a viable and sustainable protein source for ruminants. However, the small sample size limits the statistical power of the results, and further research with larger sample sizes is needed to confirm these findings. Additionally, the economic feasibility of the large-scale use of *Vachellia tortilis* should be assessed, considering factors such as availability, cost, and long-term impacts on greenhouse gaseous emissions and animal performance.

## Figures and Tables

**Figure 1 animals-15-00863-f001:**
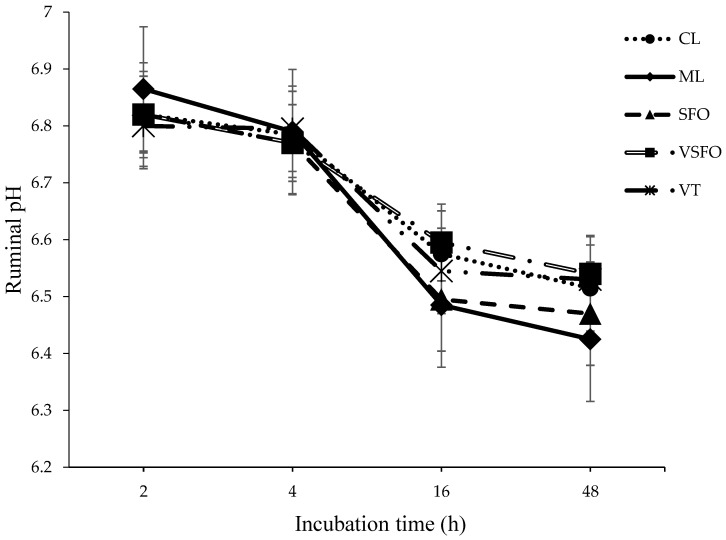
Effect of *Vachellia tortilis* leaf meal and sunflower oil on ruminal pH upon different incubation time intervals (2, 4, 16, and 48 h).

**Figure 2 animals-15-00863-f002:**
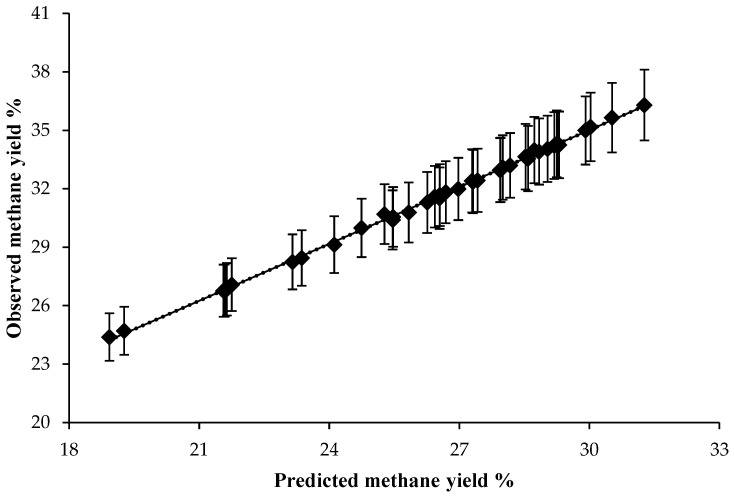
The regression relationship between molar proportions of methane calculated using equations by Wolin (1960) [26] and Moss et al. (2000) [27]. Regression line y = 1.026 (± 0.006) x − 5.916 (± 0.183). RMSE = 0.11; n = 40; adjusted R^2^ = 0.994; *p*-value = 0.0001; error bars represent standard error.

**Figure 3 animals-15-00863-f003:**
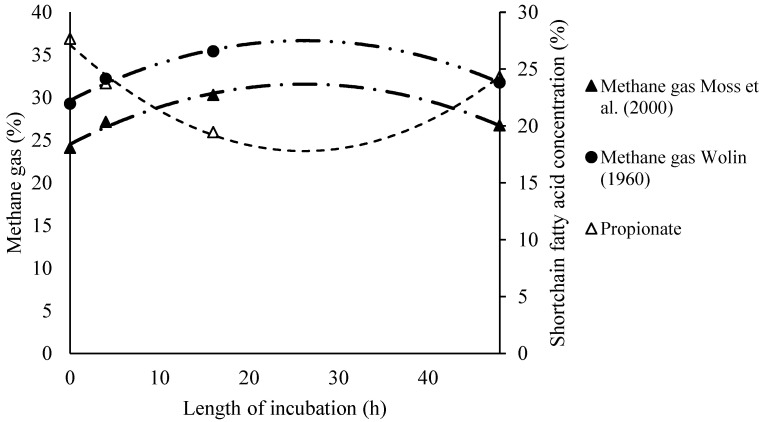
The effect of sunflower oil inclusion in concentrate supplement on the relationship of methane gas and short-chain fatty acid production with length of incubation time [28,29].

**Table 1 animals-15-00863-t001:** Ingredient inclusion levels and chemical composition (g/kg DM) of dietary treatments.

Dietary Treatments (g/kg) ^1^					
Ingredient Composition	CL	VT	VSFO	SFO	SEM
** *Vachellia tortilis* **	0.0	121.500	63.400	0.0	-
**Sunflower oil**	0.0	0.0	19.500	40.810	-
**Maize grain**	297.901	261.700	273.200	285.704	-
**Soybean meal**	212.801	186.910	195.101	204.100	-
**Lucerne**	276.601	243.0	253.700	265.300	-
**Wheat bran**	148.913	130.8	136.600	142.900	-
**Sunflower cake**	63.832	56.104	58.500	61.200	-
**Chemical Composition (g/kg) ^2^**					
**DM**	894	894	899	901	±0.001
**ASH**	54.332	67.8	61.628	55.183	±2.38
**OM**	946	932	938	945	±2.38
**CP**	229	241	257	253	±4.84
**EE**	18.401	20.3	37.802	55.8	±0.17
**NDF**	314	311	302	346	±1.17
**ADF**	143	149	145	147	±0.73
**ADL**	34	36	38	32	±0.35
**Condensed tannin**	0.0	7.101	5.672	0.0	±1.69

^1^ CL—control diet; VT—*Vachellia tortilis* leaf meal; VSFO—*Vachellia tortilis* leaf meal with sunflower oil; SFO—sunflower oil. ^2^ DM—dry matter; OM—organic matter; CP—crude protein; EE—ether extract; NDF—neutral detergent fiber; ADF—acid detergent fiber; ADL—acid detergent lignin; SEM—standard error of mean.

**Table 2 animals-15-00863-t002:** The effect of dietary treatments on individual short-chain fatty acids (SCFAs), acetate (A), propionate (P), A:P ratio, total SCFAs, methane, and carbon dioxide during incubation.

Dietary Treatments ^1^						
Parameters ^2^	CL	VT	VSFO	SFO	RMSE ^3^	*p*-Value
Total SCFAs (µmol/L)	222.237	132.053	159.264	140.580	75.789	0.161
Acetate	63.761	60.361	57.636	62.330	7.341	0.547
Propionate	23.873	24.732	25.65	23.790	3.999	0.837
Butyrate	10.749 ^b^	13.810 ^ba^	14.871 ^a^	13.870 ^ba^	2.23	0.052
A: P ratio	2.745	2.581	2.371	2.711	0.589	0.750
%CH_4_ (M)	27.072	26.346	25.570	27.061	2.656	0.778
%CH_4_ (W)	32.090	31.476	30.765	32.155	2.498	0.888
%CO_2_ (W)	56.399	58.821	60.313	57.933	3.192	0.223
IVDMD (g/kg)	44.170 ^ba^	34.333 ^b^	53.5710 ^a^	37.511 ^b^	12.458	0.026

^1^ CL—control diet; VT—*Vachellia tortilis* leaf meal diet; VSFO—*Vachellia tortilis* leaf meal and sunflower oil diet; SFO—sunflower oil diet. ^2^ SCFAs—short-chain fatty acids; A—acetate; P—propionate; CH_4_—methane; M—Moss et al. [29] equation; W—Wolin [28] equation based on using stoichiometric laws; IVDMD—in vitro dry matter digestibility; ^3^ RMSE—root mean square error; ^a,b^ means within each row with different superscripts differ significantly at *p* < 0.05.

**Table 3 animals-15-00863-t003:** Effect of incubation time intervals on total short-chain fatty acids, acetate (A), propionate (P), A:P ratio, butyrate, methane, and carbon dioxide.

Parameters ^1^	Incubation Time (h)				Multiple Comparisons		Regression Coefficient	
	2	4	16	48	RMSE ^2^	*p*-Value	Linear	Quadratic
Total SCFAs (mol/L)	153.742	199.931	162.013	134.300	71.174	0.252	−0.44 ^NS^	−0.009 ^NS^
Acetate (%)	55.120 ^b^	61.545 ^ba^	66.010 ^a^	61.7201 ^ba^	7.342	0.034	0.98 ^NS^	−0.02 **
Propionate (%)	28.302 ^a^	24.891 ^b^	21.390 ^b^	23.803 ^b^	3.399	0.002	−0.64 ^NS^	0.01 ***
Butyrate (%)	16.580	13.596	12.610	14.482	4.051	0.190	−0.33 ^NS^	0.006 ^NS^
A:P ratio	2.000 ^b^	2.566 ^ba^	3.144 ^a^	2.661 ^ba^	0.589	0.003	0.11 ^NS^	−0.002 ***
% CH_4_ (M)	23.65 ^b^	26.290 ^a^	28.857 ^a^	27.023 ^a^	2.656	0.003	0.48 *	−0.009 ***
% CH_4_ (W)	28.773 ^b^	31.352 ^a^	33.964 ^a^	32.153 ^a^	2.549	0.002	0.48 *	−0.009 ***
% CO_2_ (W)	59.500	57.382	57.261	58.533	3.262	0.390	−0.18 ^NS^	0.004 ^NS^

^1^ SCFAs—short-chain fatty acids; A—acetate; P—propionate; CH_4_—methane; M—Moss et al. [29] equation; W—Wolin [28] equation based on using stoichiometric laws; ^2^ RMSE—root mean square error. ^a,b^ means within each row with different superscripts differ significantly at *p* < 0.05; ^NS^—Non-significant at *p* > 0.05; *—significant at *p* < 0.05; **—highly significant; ***—extremely significant.

**Table 4 animals-15-00863-t004:** Quadratic regression equations of the relationship between short-chain fatty acid and methane production (Y) and length of incubation time (X).

Variables	Parameter Estimates			Regression Analysis			Turning Point		
Sunflower Oil Inclusion	Intercept	Linear	Quadratic	RMSE	R^2^-Value	*p*-Value	Type ^1^	X (h)	Y (%)
Propionate (%)	28.1 ± 1.840	−0.79 ± 2.669	0.01 ± 0.005	2.665	0.635	0.034	Min	39.50	12.50
A:P	1.9 ± 0.280	0.14 ± 0.040	−0.003 ± 0.001	0.399	0.718	0.017	Max	23.30	3.53
Methane gas production									
Sunflower oil inclusion									
% CH_4_ (M)	23.8 ± 1.410	0.59 ± 0.206	−0.01 ± 0.004	2.043	0.623	0.037	Max	29.50	32.50
% CH_4_ (W)	29.0 ± 1.380	0.59 ± 0.202	−0.01 ± 0.004	1.999	0.635	0.034	Max	29.50	37.70

A—acetate; P—propionate; CH_4_—methane; M—Moss et al. [29] equation; W—Wolin [28] equation based on using stoichiometric laws; RMSE—root mean square error; ^1^ Type, turning point is a maximum or minimum point.

**Table 5 animals-15-00863-t005:** Effect of dietary treatments on growth performance parameters of Merino lambs.

Parameters ^2^	Dietary Treatments (g/kg) ^1^					
	CL	VT	VSFO	SFO	RMSE ^3^	*p*-Value
Concentrate DMI (g DM)	433	433	431	433	0.006	0.342
Basal diet DMI (g DM)	548	583	568	534	0.081	0.245
Total DMI (g DM)	981	1016	998	968	0.083	0.233
ADG (g)	181	170	173	161	0.071	0.621
FCR (g feed/g gain)	4.10	4.80	4.90	4.70	2.594	0.457
APD (g/kg DM)	570 ^a^	480 ^ba^	440 ^b^	410 ^b^	82	0.033
ttNDFd (g/kg DM)	500	450	450	470	32	0.069

^1^ CL—control diet; VT—*Vachellia tortilis* leaf meal diet; VSFO—*Vachellia tortilis* leaf meal with sunflower oil diet; SFO—sunflower oil diet; ^2^ DMI—dry matter intake; ADG—average daily gain; FCR—feed conversion ratio; APD—apparent total tract digestibility; ttNDFd—total tract neutral detergent fiber digestibility; ^3^ RMSE—root mean square error; ^a,b^ means within each row with different superscripts differ significantly (*p* < 0.05).

## Data Availability

The data is contained within the article.

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
