# Peer review of "Effect of Vachellia tortilis Leaf Meal and Sunflower Oil Inclusion in Supplementary Diets of Lambs on In Vitro Short-Chain Fatty Acid and Gas Production and In Vivo Growth Performance"

_animals, 2025, doi:10.3390/ani15060863_

Round 1

Reviewer 1 Report

Comments and Suggestions for Authors

The study addresses relevant topics in ruminant nutrition, particularly the use of alternative feed resources and their impact on in vitro fermentation, gas production, and growth performance.
Overall, the manuscript is well-structured, with a clear objective and a logical flow of information. However, some aspects require further clarification and refinement to strengthen the scientific rigor and clarity of the findings. Below are my comments and suggestions regarding specific sections of the manuscript:
Abstract
The abstract is well-structured and provides a clear overview of the study. However, some areas could be refined, particularly in the conclusion, due to the small sample size for growth performance.
Suggestion: “However, due to the limited number of animals, further research is needed to confirm its effects on growth performance."

Introduction
1- Include more recent references (2020-2024) on methane mitigation, polyunsaturated fatty acids, and tannins in ruminant nutrition.
2- Expand the discussion on prior work involving Vachellia tortilis and sunflower oil in ruminant diets.
3- Explicitly state the novelty of the study compared to existing research.
4- Given the small number of animals, the objective should be more cautious when addressing performance evaluation. Instead of “to assess the effect on growth performance,” consider “to explore potential effects on growth performance.”

Materials and Methods
1- The study uses eight lambs divided into two live weight blocks and reassigned across periods. How were animals assigned to treatments within each period?
2- If animals switched treatments across periods, was there a carryover effect consideration? Clarify whether the periods were treated as replicates in the statistical model.
3- Given n = 2 per treatment per period, were individual animals treated as the experimental unit? If performance was analyzed, how was variability accounted for?
4- Using mixed models (PROC MIXED) instead of GLM could have been a better choice to account for repeated measures.
Results
1- Given the low number of animals (n = 2 per treatment per period), the interpretation of performance data should be more cautious.
2- Some statements suggest definitive conclusions, which should be softened (e.g., instead of saying “no negative impact,” state “no significant differences observed, though sample size was limited”).
3- Ensure consistent decimal places across all values.
4- Clearly separate groups (treatments, time points, etc.) for better readability.
5- The effects of Vachellia tortilis and sunflower oil on SCFA production and gas emissions could be better explained in relation to previous literature.
6- Instead of just reporting numbers, briefly explain their biological significance.

Discussion
1- The discussion effectively explains the results, but some interpretations could be more precise. For example, the impact of Vachellia tortilis on fermentation should be compared more directly with previous studies. While some references are cited, a stronger connection between findings and literature is needed.
2- The methane and carbon dioxide results indicate no significant differences, but the discussion does not fully explore why this might be the case. Was it due to the fiber composition, tannins, or another factor? Including potential mechanisms would strengthen the interpretation.
3- The butyrate increase with VT and VSFO is noted but needs better explanation. What is the possible biological significance of this shift in fermentation? Does it suggest a higher energy yield for the animal, or a change in microbial populations?
4- Growth performance results must be discussed with caution. Given the small sample size, any trends observed should be described as preliminary findings rather than conclusive evidence. Instead of saying that VT had no effect, it should be framed as "no significant effect was detected, but further research is needed to confirm this."
5- The discussion should highlight study limitations more explicitly. The small number of animals limits the statistical power for detecting differences in performance. This should be clearly acknowledged before making recommendations for practical application.

Conclusions
1- The conclusion effectively summarizes the findings but could be more structured and explicit about the study's limitations. Given the small number of animals, the statement regarding growth performance should be more cautious to avoid overgeneralization.
2- The conclusion should follow a logical order, first summarizing the fermentation and gas production findings, then addressing performance, and finally discussing the practical implications of using Vachellia tortilis.
3- The economic feasibility of using Vachellia tortilis should be mentioned. While it appears to be a viable protein source, does its availability and cost justify large-scale use?
4- The conclusion should explicitly acknowledge study limitations (sample size, experimental design constraints) and suggest directions for future research.
All citations in the text are properly listed in the reference section. However, there are references in the list that are not cited in the manuscript.

Author Response

Reviewer 1

Open Review

(x) I would not like to sign my review report

( ) I would like to sign my review report

Quality of English Language

(x) The English is fine and does not require any improvement.

( ) The English could be improved to more clearly express the research.

Yes         Can be improved            Must be improved          Not applicable

Does the introduction provide sufficient background and include all relevant references?

( )            ( )            (x)           ( )

Is the research design appropriate?

( )            (x)           ( )            ( )

Are the methods adequately described?

(x)           ( )            ( )            ( )

Are the results clearly presented?

( )            (x)           ( )            ( )

Are the conclusions supported by the results?

( )            (x)           ( )            ( )

Comments and Suggestions for Authors

The study addresses relevant topics in ruminant nutrition, particularly the use of alternative feed resources and their impact on in vitro fermentation, gas production, and growth performance.

Overall, the manuscript is well-structured, with a clear objective and a logical flow of information. However, some aspects require further clarification and refinement to strengthen the scientific rigor and clarity of the findings. Below are my comments and suggestions regarding specific sections of the manuscript:

Abstract

The abstract is well-structured and provides a clear overview of the study. However, some areas could be refined, particularly in the conclusion, due to the small sample size for growth performance.

Suggestion: “However, due to the limited number of animals, further research is needed to confirm its effects on growth performance."

Introduction

Reviewer one

Introduction

1- Include more recent references (2020-2024) on methane mitigation, polyunsaturated fatty acids, and tannins in ruminant nutrition.

Addressed in the manuscript

2- Expand the discussion on prior work involving Vachellia tortilis and sunflower oil in ruminant diets.

Addressed in the manuscript

3- Explicitly state the novelty of the study compared to existing research.

Addressed in the manuscript

4- Given the small number of animals, the objective should be more cautious when addressing performance evaluation. Instead of “to assess the effect on growth performance,” consider “to explore potential effects on growth performance.”

Addressed in the manuscript

Materials and methods

1- The study uses eight lambs divided into two live weight blocks and reassigned across periods. How were animals assigned to treatments within each period?

Animals were initially blocked by live weight into two groups to control for weight-related variability. Within each block, lambs were randomly assigned to one of the four dietary treatments (Control, VT, SFO, VSFO) for Period 1. In subsequent periods, treatments were reassigned in a balanced manner so that each animal received all treatments once over the three periods. This ensured that no treatment was immediately repeated for the same animal, minimizing bias.

2- If animals switched treatments across periods, was there a carryover effect consideration? Clarify whether the periods were treated as replicates in the statistical model.

Since animals switched treatments across periods, potential carryover effects (previous diet influencing subsequent performance) could arise. To mitigate this, a washout period or an adaptation phase of 14 days at the start of each period was introduced to help minimise residual effects.

Periods were treated as replicates; the statistical model include Period as a fixed/repeated effect and Animal within Block as a random effect to account for repeated measures

3- Given n = 2 per treatment per period, were individual animals treated as the experimental unit? If performance was analyzed, how was variability accounted for?

individual animals (n = 2 per treatment per period) were treated as the experimental unit, as each lamb’s response to the diet was measured independently.

Variability was accounted for by:

Blocking by weight, reducing variation due to initial size differences.

Including animal as a random effect in the statistical model to control for individual differences.

Repeated measures analysis, which helps account for within-animal variation across periods

4- Using mixed models (PROC MIXED) instead of GLM could have been a better choice to account for repeated measures.

Results

Proc mixed was used to re-analyse the data that was originally analysed using prof GLM of SAS and the was a typographic error in the revised version of the manuscript where authors omitted deleting prof GLM

1- Given the low number of animals (n = 2 per treatment per period), the interpretation of performance data should be more cautious.

Noted with thanks

2- Some statements suggest definitive conclusions, which should be softened (e.g., instead of saying “no negative impact,” state “no significant differences observed, though sample size was limited”).

Addressed in the manuscript

3- Ensure consistent decimal places across all values.

Addressed in the manuscript

4-Clearly separate groups (treatments, time points, etc.) for better readability

Addressed in the manuscript according to our understanding as the comment is not clear

5- The effects of Vachellia tortilis and sunflower oil on SCFA production and gas emissions could be better explained in relation to previous literature.

Addressed in the manuscript

6- Instead of just reporting numbers, briefly explain their biological significance.

Addressed in the results section of the manuscript

Discussion

1- The discussion effectively explains the results, but some interpretations could be more precise. For example, the impact of Vachellia tortilis on fermentation should be compared more directly with previous studies. While some references are cited, a stronger connection between findings and literature is needed.

2- The methane and carbon dioxide results indicate no significant differences, but the discussion does not fully explore why this might be the case. Was it due to the fiber composition, tannins, or another factor? Including potential mechanisms would strengthen the interpretation.

3- The butyrate increase with VT and VSFO is noted but needs better explanation. What is the possible biological significance of this shift in fermentation? Does it suggest a higher energy yield for the animal, or a change in microbial populations?

4- Growth performance results must be discussed with caution. Given the small sample size, any trends observed should be described as preliminary findings rather than conclusive evidence. Instead of saying that VT had no effect, it should be framed as "no significant effect was detected, but further research is needed to confirm this."

5- The discussion should highlight study limitations more explicitly. The small number of animals limits the statistical power for detecting differences in performance. This should be clearly acknowledged before making recommendations for practical application.

Comments incorporated in the revised version of the manuscript

Conclusions

1- The conclusion effectively summarizes the findings but could be more structured and explicit about the study's limitations. Given the small number of animals, the statement regarding growth performance should be more cautious to avoid overgeneralization.

2- The conclusion should follow a logical order, first summarizing the fermentation and gas production findings, then addressing performance, and finally discussing the practical implications of using Vachellia tortilis.

3- The economic feasibility of using Vachellia tortilis should be mentioned. While it appears to be a viable protein source, does its availability and cost justify large-scale use?

4- The conclusion should explicitly acknowledge study limitations (sample size, experimental design constraints) and suggest directions for future research.

All citations in the text are properly listed in the reference section. However, there are references in the list that are not cited in the manuscript.

Comments are addressed in the manuscript

Reviewer 2

Comments and Suggestions for Authors

In the manuscript, the authors studied the Effect of Vachellia tortilis leaf meal and sunflower oil inclusion on supplementary diets of lambs on in vitro short-chain fatty acid production and growth performance.
However, the following comments can be made.
1. Research methodology. Why is the number of animals in one study group 2? Is this enough to identify a natural effect?
2. Research methodology. In what connection was the use of sunflower oil studied in the study? How is this related to this study?
3. Is the studied plant cultivated in this region? What is the annual volume of its cultivation. Is this volume sufficient for its use as animal feed?
4. I do not see the point in using in vitro acid assessment. The obtained rumen fluid could have been immediately analyzed for the concentration of the studied acids! In this case, the studies would have been in vivo and would have been more important.
5. Results. It would be desirable to show the chemical composition or nutritional value of the studied plant.
6. Results. Tables 2-4 - no mean error.
7. Conclusion. There is no assessment and analysis of the results obtained. It is written very briefly and not clearly.

Comments effected

Reviewer 3

Experimental design:

  • The sample size of eight lambs (two per treatment) is small and may not provide sufficient statistical power for conclusive results, despite three experimental repetitions. Justification should be provided by citing articles that have used a similar sample size and replication method.
  • In the discussion, the authors state, "Our results for VT and VSFO diets had a higher IVDMD compared to the control diet." However, this claim is incorrect based on the reported results. The authors should revise this statement to align with the data presented.
  •  

Additional Points Requiring Clarification:

  • The chemical characterization of Vachellia tortilis leaf meal should be presented in detail to provide better context for its nutritional contribution.
  • The feed intake measurement should specify whether intake was measured daily or at specific intervals.
  • Ensure that all citations follow the correct journal formatting style.

Tables:

  • Table 3 contains multiple comparisons, but the significant differences are not clearly highlighted. Consider using superscripts or bold text to make these distinctions more visible.
  • The format of tables should be standardized to improve readability and consistency.

Comments addressed in the revised version of the manuscript

Reviewer 2 Report

Comments and Suggestions for Authors

In the manuscript, the authors studied the Effect of Vachellia tortilis leaf meal and sunflower oil inclusion on supplementary diets of lambs on in vitro short-chain fatty acid production and growth performance.
However, the following comments can be made.
1. Research methodology. Why is the number of animals in one study group 2? Is this enough to identify a natural effect?
2. Research methodology. In what connection was the use of sunflower oil studied in the study? How is this related to this study?
3. Is the studied plant cultivated in this region? What is the annual volume of its cultivation. Is this volume sufficient for its use as animal feed?
4. I do not see the point in using in vitro acid assessment. The obtained rumen fluid could have been immediately analyzed for the concentration of the studied acids! In this case, the studies would have been in vivo and would have been more important.
5. Results. It would be desirable to show the chemical composition or nutritional value of the studied plant.
6. Results. Tables 2-4 - no mean error.
7. Conclusion. There is no assessment and analysis of the results obtained. It is written very briefly and not clearly.

Author Response

(The authors gave the same response as above.)

Reviewer 3 Report

Comments and Suggestions for Authors

General comments:
The manuscript investigates the impact of Vachellia tortilis leaf meal and sunflower oil supplementation on in vitro short-chain fatty acid production, gas production, and growth performance of lambs. The research addresses an important topic in animal nutrition, but certain areas require clarification.

Experimental design:

  • The sample size of eight lambs (two per treatment) is small and may not provide sufficient statistical power for conclusive results, despite three experimental repetitions. Justification should be provided by citing articles that have used a similar sample size and replication method.
  • In the discussion, the authors state, "Our results for VT and VSFO diets had a higher IVDMD compared to the control diet." However, this claim is incorrect based on the reported results. The authors should revise this statement to align with the data presented.
  •  

Additional Points Requiring Clarification:

  • The chemical characterization of Vachellia tortilis leaf meal should be presented in detail to provide better context for its nutritional contribution.
  • The feed intake measurement should specify whether intake was measured daily or at specific intervals.
  • Ensure that all citations follow the correct journal formatting style.

Tables:

  • Table 3 contains multiple comparisons, but the significant differences are not clearly highlighted. Consider using superscripts or bold text to make these distinctions more visible.
  • The format of tables should be standardized to improve readability and consistency.

Author Response

(The authors gave the same response as above.)

Reviewer 4 Report

Comments and Suggestions for Authors

The entire manuscript requires significant revisions.

  • The title should be revised.
  • Some spelling and grammatical corrections should be made in the abstract.
  • References should be provided for the information given in the introduction.
  • The introduction discusses the importance of alternative feed sources considering low nutritional value, methane gas emissions, limited resources, high costs, and climate conditions. However, the feed materials used in the study are not directly related to these descriptions. Therefore, the study's objectives and methods should be more clearly linked.
  • In the Materials and Methods section, the study area and the species present in this area are mentioned. However, it is stated that one of the feed additives (VT) used in the study was harvested from a different area. The reason for using a different area and its sustainability should be justified. The reason for using the other feed additive (SFO) is not specified.
  • The abbreviation for the control diet (CT) used in the Materials and Methods section is inconsistently referred to as (CL) in other sections. The same abbreviation should be used consistently throughout the manuscript.
  • The VT:SFO ratios in the CSFO diet mentioned in the Materials and Methods section should be provided.
  • The experimental design in the Materials and Methods section should be explained more clearly. Having only two lambs per group raises concerns regarding the statistical reliability of the results. The reliability of the results should be justified. Additionally, it is unclear how four different treatments were applied across three periods. The reason for using different periods should be explained. The assumption that group switching between periods does not influence results should be justified. The reliability of this method should be supported with references.
  • Detailed information should be provided on where the lambs were kept throughout the study.
  • In the Materials and Methods section, the subsection titled “in vitro digestion” should be revised to specify the use of rumen fluid. Additionally, details on how the rumen fluid was obtained should be included.
  • References should be provided for the analytical methods described in the Materials and Methods section.
  • The results section should be rewritten, considering the highlighted points. Incorrect and unclear statements should be corrected.
  • Data in the results section should be presented in an organized manner. Each table and topic should be discussed in separate paragraphs, and unrelated topics should not be mentioned in the same paragraph.
  • Although pH values are not included in the study, they are frequently mentioned in the results and discussion sections. The pH data should be added to the relevant table.
  • The number of decimal places in tables should be consistent after the decimal point. The p-value should be reported with at least three decimal places.
  • There is almost no discussion or interpretation of Table 4. If the table does not contribute to the study, it should be removed entirely.
  • Table 5 only presents data related to SFO inclusion. The reason for this should be explained, or data related to VT supplementation should be included similarly. Additionally, the results and discussion of this table should be elaborated in more detail.
  • In Table 6, the initial and final body weights necessary for the growth performance table should be provided. The similarity of DMI data is concerning, yet the differences between groups are reported as highly significant. This may be due to the small sample size. Additionally, low error values could be evidence of this issue.

Comments on the Quality of English Language

The focus of the study and the value given to scientific knowledge are truly commendable. However, the manuscript requires major revisions in terms of its presentation and certain scientific aspects.

In general, the sections that need revision have been highlighted in the manuscript below. Additionally, all details have been marked and commented on in the PDF file. This has been done with the utmost respect and courtesy.

  • The title should be revised.
  • Some spelling and grammatical corrections should be made in the abstract.
  • References should be provided for the information given in the introduction.
  • The introduction highlights the importance of alternative feed sources in the context of low nutritional value, methane gas emissions, limited resources, high costs, and climate conditions. However, the feed materials used in the study are not directly related to these descriptions. Therefore, the study’s objectives and methods should be more clearly aligned.
  • In the Materials and Methods section, the study area and the species present in this area are mentioned. However, one of the feed additives (VT) used in the study was harvested from a different area. The reason for using a different area and its sustainability should be justified. The reason for using the other feed additive (SFO) is not specified.
  • The abbreviation for the control diet (CT) used in the Materials and Methods section is inconsistently referred to as (CL) in other sections. The same abbreviation should be used consistently throughout the manuscript.
  • The VT:SFO ratios in the CSFO diet mentioned in the Materials and Methods section should be provided.
  • The experimental design in the Materials and Methods section should be explained more clearly. Having only two lambs per group raises concerns regarding the statistical reliability of the results. The reliability of the results should be justified. Additionally, it is unclear how four different treatments were applied across three periods. The reason for using different periods should be explained. The assumption that group switching between periods does not influence results should be justified. The reliability of this method should be supported with references.
  • Detailed information should be provided on where the lambs were kept throughout the study.
  • In the Materials and Methods section, the subsection titled “in vitro digestion” should be revised to specify the use of rumen fluid. Additionally, details on how the rumen fluid was obtained should be included.
  • References should be provided for the analytical methods described in the Materials and Methods section.
  • The results section should be rewritten, considering the highlighted points. Incorrect and unclear statements should be corrected.
  • Data in the results section should be presented in an organized manner. Each table and topic should be discussed in separate paragraphs, and unrelated topics should not be mentioned in the same paragraph.
  • Although pH values are not included in the study, they are frequently mentioned in the results and discussion sections. The pH data should be added to the relevant table.
  • The number of decimal places in tables should be consistent after the decimal point. The p-value should be reported with at least three decimal places.
  • There is almost no discussion or interpretation of Table 4. If the table does not contribute to the study, it should be removed entirely.
  • Table 5 only presents data related to SFO inclusion. The reason for this should be explained, or data related to VT supplementation should be included similarly. Additionally, the results and discussion of this table should be elaborated in more detail.
  • In Table 6, the initial and final body weights necessary for the growth performance table should be provided. The similarity of DMI data is concerning, yet the differences between groups are reported as highly significant (P = 0.0001). This may be due to the small sample size. Additionally, the low error value (RMSE) could also be a contributing factor.
  • In Table 6, further clarification is needed regarding the ttNDFd parameter…

Author Response

(The authors gave the same response as above.)

Round 2

Reviewer 3 Report

Comments and Suggestions for Authors

The sample size of eight lambs, with only two animals per treatment group, is insufficient and limits the statistical power of the study, even with three experimental repetitions. Such a small sample size undermines the reliability and generalizability of the findings. To strengthen the study’s validity, it is recommended that a larger sample size be used. If repeating the study with a larger sample is not feasible, the authors should cite relevant studies that have employed similar methodologies to support their approach.

 in Table 2, the concentrations of SCFAs are reported, but the units are not specified

Author Response

Response to reviewers

Reviewer 3

The sample size of eight lambs, with only two animals per treatment group, is insufficient and limits the statistical power of the study, even with three experimental repetitions. Such a small sample size undermines the reliability and generalizability of the findings. To strengthen the study’s validity, it is recommended that a larger sample size be used. If repeating the study with a larger sample is not feasible, the authors should cite relevant studies that have employed similar methodologies to support their approach.

Here are the cited studies with the methodology aligning with  the experimental design (number of animals) used in the current study  “reference 20-23”

in Table 2, the concentrations of SCFAs are reported, but the units are not specified

The comment is addressed in Table 2 (µmol/L)

Reviewer 4

The title should be revised.

Tittle revised from “Effect of Vachellia tortilis leaf meal and sunflower oil inclusion on supplementary diets of lambs on in vitro short-chain fatty production and growth performance” to “Effect of Vachellia tortilis leaf meal and sunflower oil inclusion in supplementary diets of lambs on in vitro short-chain fatty acid and gas production, and in vivo growth performance

Some spelling and grammatical corrections should be made in the abstract.

Comment addressed throughout the manuscript

References should be provided for the information given in the introduction.

Introduction improved and relevant references used to address the importance of incorporating Vachellia tortilis (reference 10 and 11) and sunflower oils (reference 12 and 13)

The introduction discusses the importance of alternative feed sources considering low nutritional value, methane gas emissions, limited resources, high costs, and climate conditions. However, the feed materials used in the study are not directly related to these descriptions. Therefore, the study's objectives and methods should be more clearly linked.

Line 78-87 spells out the importance of the materials used and backed up with the objective in line 88-90

The methodology is improved to capture the objectives of the study.

In the Materials and Methods section, the study area and the species present in this area are mentioned. However, it is stated that one of the feed additives (VT) used in the study was harvested from a different area. The reason for using a different area and its sustainability should be justified. The reason for using the other feed additive (SFO) is not specified.

The study area is in the same province with the area where VT leaves were harvested. It is noteworthy to mention that the study was conducted in a research station in the Institution not in the rangelands

Line 80-82 explains the use of SFO.

The abbreviation for the control diet (CT) used in the Materials and Methods section is inconsistently referred to as (CL) in other sections. The same abbreviation should be used consistently throughout the manuscript.

The error is acknowledged and was corrected

The VT:SFO ratios in the CSFO diet mentioned in the Materials and Methods section should be provided

Inclusion levels of dietary ingredients are included in Table 1

The experimental design in the Materials and Methods section should be explained more clearly. Having only two lambs per group raises concerns regarding the statistical reliability of the results. The reliability of the results should be justified. Additionally, it is unclear how four different treatments were applied across three periods. The reason for using different periods should be explained. The assumption that group switching between periods does not influence results should be justified. The reliability of this method should be supported with references.

Animals were initially blocked by live weight into two groups to control for weight-related variability. Within each block, lambs were randomly assigned to one of the four dietary treatments (Control, VT, SFO, VSFO) for Period 1. In subsequent periods, treatments were reassigned in a balanced manner so that each animal received all treatments once over the three periods. This ensured that no treatment was immediately repeated for the same animal, minimizing bias.

Since animals switched treatments across periods, potential carryover effects (previous diet influencing subsequent performance) could arise. To mitigate this, a washout period or an adaptation phase of 14 days at the start of each period was introduced to help minimise residual effects.

Periods were treated as replicates; the statistical model include Period as a fixed/repeated effect and Animal within Block as a random effect to account for repeated measures.   individual animals (n = 2 per treatment per period) were treated as the experimental unit, as each lamb’s response to the diet was measured independently. Also, blocking by weight, reducing variation due to initial size differences.

Including animal as a random effect in the statistical model to control for individual differences.

Repeated measures analysis, which helps account for within-animal variation across periods

Detailed information should be provided on where the lambs were kept throughout the study.

Addressed in the previous response

In the Materials and Methods section, the subsection titled “in vitro digestion” should be revised to specify the use of rumen fluid. Additionally, details on how the rumen fluid was obtained should be included.

Line 191-197 specify the method of rumen fluid collection in the manuscript

References should be provided for the analytical methods described in the Materials and Methods section.

Procedures were referenced where applicable

The results section should be rewritten, considering the highlighted points. Incorrect and unclear statements should be corrected.

Comment effected

Data in the results section should be presented in an organized manner. Each table and topic should be discussed in separate paragraphs, and unrelated topics should not be mentioned in the same paragraph.

The results were mingled in the discussion section to facilitate the link amongst the results of each Table or figure.

Although pH values are not included in the study, they are frequently mentioned in the results and discussion sections. The pH data should be added to the relevant table.

The figure with pH values was included in the manuscript

The number of decimal places in tables should be consistent after the decimal point. The p-value should be reported with at least three decimal places.

Comment acknowledged and addressed

There is almost no discussion or interpretation of Table 4. If the table does not contribute to the study, it should be removed entirely.

Table 4 was removed, as there were no observed significant differences amongst the findings.  

Table 5 only presents data related to SFO inclusion. The reason for this should be explained, or data related to VT supplementation should be included similarly. Additionally, the results and discussion of this table should be elaborated in more detail.

Only data that showed significant findings were presented on some of the observed parameters of interest

In Table 6, the initial and final body weights necessary for the growth performance table should be provided. The similarity of DMI data is concerning, yet the differences between groups are reported as highly significant. This may be due to the small sample size. Additionally, low error values could be evidence of this issue.

The mean initial body weight was 27.15 ± 3.0 kg, across treatments.

Reviewer 4 Report

Comments and Suggestions for Authors

The requested revisions were not properly addressed. In fact, there was a lack of changes.

Author Response

(The authors gave the same response as above.)
